# Sparse is Enough in Scaling Transformers

**Sebastian Jaszczur**[*]  **Aakanksha Chowdhery**  **Afroz Mohiuddin**  **Łukasz Kaiser**[*]
University of Warsaw    Google Research    Google Research    OpenAI

**Wojciech Gajewski**     **Henryk Michalewski**     **Jonni Kanerva**
Google Research        Google Research        Google Research

## Abstract

Large Transformer models yield impressive results on many tasks, but are expensive to train, or even fine-tune, and so slow at decoding that their use and study becomes out of reach. We address this problem by leveraging sparsity. We study sparse variants for all layers in the Transformer and propose *Scaling Transformers*, a family of next generation Transformer models that use sparse layers to scale efficiently and perform unbatched decoding much faster than the standard Transformer as we scale up the model size. Surprisingly, the sparse layers are enough to obtain the same perplexity as the standard Transformer with the same number of parameters. We also integrate with prior sparsity approaches to attention and enable fast inference on long sequences even with limited memory. This results in performance competitive to the state-of-the-art on long text summarization.

## 1 Introduction

The field of natural language processing has seen dramatic improvements in recent years due to large neural networks based on the Transformer architecture. The original Transformer [42] significantly advanced state-of-the-art in machine translation. BERT [7] surpassed all previous methods on question answering, language inference and other NLP tasks and was followed by a line of models like T5 [30] that further improved these results. The GPT line of models [29, 3] elevated language generation to the point that GPT-2 was invited to write short passages for the Economist and GPT-3 created whole articles almost indistinguishable from human-written ones.

The benefits of this progress are undercut by the huge costs such models incur. Strubell et al. [36] estimate that training a single base BERT model costs \$4k-\$12k and emits as much $CO_2$ as one passenger's share of a 4-hour flight and later Patterson et al. [27] estimate that training GPT-3 has three times as much $tCO_2e$ (metric tons of $CO_2$ equivalent) emissions as a SF-NY round trip flight. Data and serving costs are also forbidding: a single training run of BERT, for example, processes 128B tokens, and Google Translate reportedly[1] serves over 143B words per day.

With the growing popularity and size of these models, it is increasingly valuable to make them scale efficiently. In this work we propose *Scaling Transformers* with a separate *sparse mechanism for the query, key, value and output layers* (QKV layers for short) and combine it with *sparse feedforward blocks* to get a fully sparse Transformer architecture.

To quantify the computational complexity of inference in Transformer models, recall the architecture of a Transformer decoder block. It consists of three parts: a masked self-attention layer, an encoder-decoder attention layer and a feedforward block. The sizes of these layers are parameterized by $d_{model}$ and $d_{ff}$. The base BERT model sets $d_{model} = 768$, the large BERT has $d_{model} = 1024$, the largest

---

[*]Work done while at Google Research.
[1]`https://cutt.ly/skkFJ7a`

35th Conference on Neural Information Processing Systems (NeurIPS 2021).

| | Params | Dec. time | Dec. time per block |
|---|---|---|---|
| baseline Transf. | 800M | 0.160s | 5.9ms |
| + Sparse FF | - | 0.093s | 3.1ms |
| + Sparse QKV | - | 0.152s | 6.2ms |
| + Sparse FF+QKV | - | 0.061s | 1.9ms |
| Speedup | | 2.62x | 3.05x |
| baseline Transf. | 17B | 3.690s | 0.581s |
| +Sparse FF | - | 1.595s | 0.259s |
| +Sparse QKV | - | 3.154s | 0.554s |
| +Sparse FF+QKV | - | 0.183s | 0.014s |
| Speedup | | 20.0x | 42.5x |

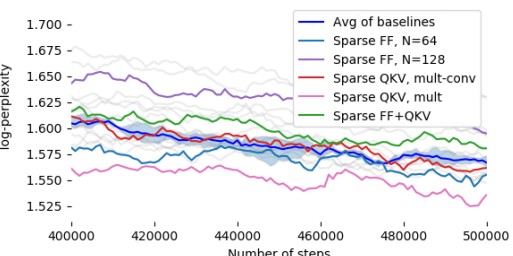

*Table 1: Decoding speed (in seconds) of a single token. For Transformer model (equivalent to T5 large with approximately 800M parameters), Scaling Transformers with proposed sparsity mechanisms (FF+QKV) achieve up to 2x speedup in decoding compared to baseline dense model and 20x speedup for 17B param model.*

*Figure 1: Log-perplexity of Scaling Transformers (equivalent to T5 large with approximately 800M parameters) on C4 dataset with proposed sparsity mechanisms (FF, QKV, FF+QKV) is similar to baseline dense model. Other models used in this paper are shown in grey lines; raw data is available in the appendix.*

GPT-2 has $d_{\text{model}} = 1600$ and GPT-3 reaches $d_{\text{model}} = 12288$. For both BERT and GPT models the authors use $d_{\text{ff}} = 4\,d_{\text{model}}$. While decoding a token, the self-attention layer needs to activate four matrices of size $d_{\text{model}} \times d_{\text{model}}$: one each for the queries, keys and values input to the attention and one for merging the output. In the encoder-decoder attention, the keys and values may already be cached, so only two matrices of size $d_{\text{model}} \times d_{\text{model}}$ are activated. The feedforward block consists of two matrices of size $d_{\text{model}} \times d_{\text{ff}}$, omitting small additional contribution of biases. The total adds up to: $4\,d_{\text{model}}^2 + 2\,d_{\text{model}}^2 + 2\,d_{\text{model}}\,d_{\text{ff}}$. This sum describes both the number of trainable weights of a single block and approximates well the number of floating-point operations needed for decoding a single token, except for the attention operations (discussed later). The complexity is quadratic in $d_{\text{model}}$; for example, as $d_{\text{model}}$ increases 16-fold from base BERT to GPT-3, the complexity of a single block grows 256-fold.

In comparison *Scaling Transformers* use only $2d_{\text{model}}\sqrt{d_{\text{model}}} = 2d_{\text{model}}^{1.5}$ parameters in QKV layers and yield results as good as the baseline (fully dense) Transformer with the same number of parameters and complexity: $8\,d_{\text{model}}^{1.5} + 4\,d_{\text{model}}^{1.5} + 4\,d_{\text{model}}^{1.5}$. We were surprised that the fully sparse *Scaling Transformers* are indeed enough to match the results of the baseline Transformer on the large C4 dataset [30] (Figure 1). The improvement in complexity holds not just asymptotically but yields over 2.6x speedup in wall-clock hed decoding time already for a model with 800M parameters and 20x improvement for a model with 17B parameters, as shown in Table 1.

To verify that Scaling Transformers can be used with other Transformer improvements on real tasks, we create *Terraformer* – a Transformer model that uses reversible layers for memory efficiency and sparse attention to handle long sequences. We pre-train Terraformer on the C4 dataset and fine-tune it on the challenging task of summarizing arxiv articles. Terraformer yields results competitive to the state-of-the-art BigBird-Pegasus without using the Pegasus loss in pre-training (Table 5).

## 2   Related Work

As discussed in the previous section, large Transformer models brings significant improvements in performance, as seen in models such as GPT-3 [3, 17] or T5 [44, 30]. Training and inference incur a high computational cost at the scale of hundreds of billions of parameters. Numerous techniques improve the efficiency of Transformer models, and Gupta and Agrawal [11] divide them into several classes, including pruning, knowledge distillation, quantization, parameter sharing, efficient attention, and efficient feedforward.

**Model compression.** Model pruning [24, 2] makes matrices smaller by removing unneeded weights after or during training, however, the gains in computational complexity on sparse matrices often do

---

[2]The 800M model has 24 layers of Encoder & Decoder, $d_{\text{model}} = 1024$, 16 attn heads, attention-sparsity = 16, ff-sparsity = 64. We scale up this model to approximately 17B parameters with $d_{\text{model}} = 9216$ and get up to 20x speedup in decoding compared to baseline dense model. This 17B param model has six layers of Encoder & Decoder, 96 attn heads, attention-sparsity = 64, ff-sparsity = 256.

not result in inference speedups on actual hardware [9]. Structured pruning based approaches [47, 22, 43] account for this challenge by leveraging sparsity in hardware in CPU and GPU architectures [1]. Our paper is different from pruning approaches in that it relies on dynamic sparsity wherein the feedforward layer loads only a subset of weights in the layer for each token. Our approach is complementary to model quantization studies [35, 38, 28] that use fewer bits for the weights.

**Model distillation.** Several natural language models used for mobile inference [13, 39] rely on distillation [32] to speed up inference from the pretrained large models. For example, [18] pretrains a large model and uses knowledge distillation along with pruning to get more than 10x faster inference. Instead of distilling a large model, our approach speeds up inference by reducing the number of weights loaded in memory from the model.

**Sparse attention.** Sparse attention-based approaches have made the attention layer more efficient, especially for long sequences, by incorporating additional combinatorial mechanisms, as in [40], or selecting a subset of tokens this layer attends to [31, 5, 19, 37, 15, 4] or other approaches [12]. Our work is complementary to these approaches for sparse attention and reuses the advances on SOTA therein. Inference speedups in the attention layers also use bottleneck layers [39] or grouped convolutions [13]. Our work extends beyond the idea of grouped convolutions approach because each attention head is limited to using only a fixed part of the embedding while our work is able to permute the embeddings to improve model quality; see Section 3.2 for details.

**Tensor Decomposition.** The approaches discussed above significantly improve Transformer speed and handling of long sequences, however none of them addresses the fundamental scaling issue: even if we distill into a smaller model, quantize it and prune a percentage of the weights, the complexity still grows quadratically with $d_{model}$. The final approach, which does attack this scaling issue, is called *tensor decompositions* in [11]. Unluckily, as the authors there note, the approach is most effective in dealing with large input and output embedding matrices and tends to produce lower performance than unstructured models if used inside the decoder block.

**Sparse feedforward.** Mixture of experts approaches have been shown to achieve computational efficiency in training [33, 21, 34], scaling up to a trillion parameters [8]. The key idea is to partition the $d_{ff}$-sized dimension into parts (called experts) and retrieve only one part per token, which reduces the complexity of the feedforward block from $2d_{model}d_{ff}$ to $2d_{model}d_{ff}/n_{experts}$. These speedups are mostly measured in training speed, and the method focuses on feedforward blocks. In contrast to prior methods, we train a full weight matrix and then only activate specific parts of it for each input token during decoding; see Section 3.1.

## 3   Sparse is Enough

We study how to sparsify every part of the Transformer model—otherwise the non-sparse parts dominate decoding time and become a bottleneck. This means we need sparse equivalents for the feedforward blocks, for the dense Q, K, V and output layers in attention, and for the final dense layer before the softmax and loss.

### 3.1   Sparse Feedforward Layer

In a baseline Transformer, decoding speed is dominated by the execution cost of the feedforward block. Recall that this block consists of two fully-connected (dense) layers with a ReLU nonlinearity in between. The dimensionality of activation vectors between these 2 layers is usually denoted by $d_{ff}$ and is often 4 or 8 times larger than the dimensionality of the activations in other places ($d_{model}$).

We make use of the structure of the feedforward block to sparsify it. One main observation is that the ReLU in the middle creates a lot of zeros[2]. We impose a fixed structure on this middle activation vector: only one float in every block of $N$ will be allowed to be non-zero. Prior techniques prune weights or blocks from weight matrices and can be referred to as static sparsity. Our proposed technique will train a full weight matrix but only activate specific parts of it for each input token during decoding. We call this dynamic sparsity, because the model dynamically selects only a fraction of its parameters, and the selection is independent for each token.

---

[2]GeLU is another non-linearity often used in the Transformer feedforward block. Table 1 in [26] shows the same final loss for ReLU and GeLU on the C4 dataset, though, so in this work for simplicity, we focus on ReLU.

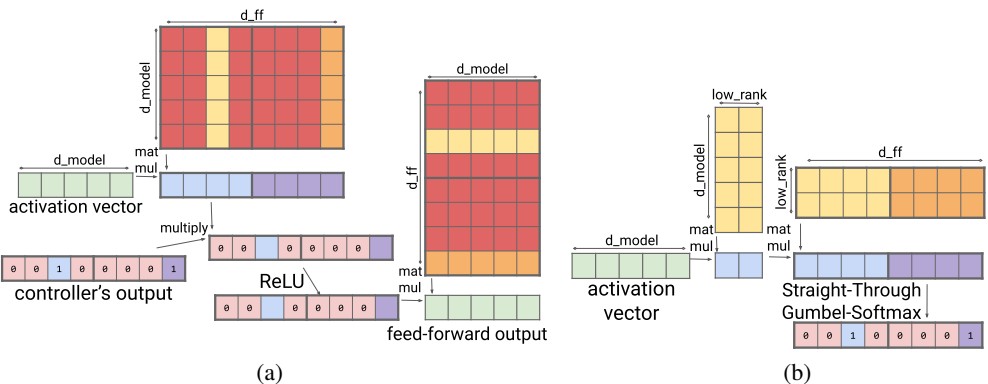

Figure 2: (a) Sparse Feedforward Layer only activates 1 in N rows/columns of each block to reduce the decoding time. Here only two rows/colums in blocks of size 4 are loaded while the weights in dark red are not loaded from memory during inference. (b) Sparse Feedforward Controller with the output of 2 blocks of size 4 (1 in 4 sparsity).

We train a controller to determine which activation in each block can be non-zero; the rest will be set to zero. This can be represented as

$$Y_{\text{sparse}} = \max(0, xW_1 + b_1) \odot \text{Controller}(x)$$
$$\text{SparseFFN}(x) = Y_{\text{sparse}}W_2 + b_2$$

where $\odot$ is element-wise multiplication. Note that each activation in $Y_{\text{sparse}}$ corresponds to a single column in $W_1$ and a single row in $W_2$. Therefore, if we compute Controller($x$) output first, we don't have to use any columns in $W_1$ or any rows in $W_2$ that correspond to an activation set to zero by the controller. This allows for much faster decoding, as we have to process only 1 in $N$ columns in $W_1$ and rows in $W_2$ (see Figure 2(a)).

To design the controller to be computationally inexpensive, we project the input using a low-rank bottleneck dense layer. Figure 2(b) illustrates the controller which produces the output as follows

$$\text{Controller}(x) = \arg\max(\text{Reshape}(xC_1C_2, (-1, N)))$$

where $C_1 \in \mathbb{R}^{d_{\text{model}} \times d_{\text{lowrank}}}$ and $C_2 \in \mathbb{R}^{d_{\text{lowrank}} \times d_{\text{ff}}}$, with $d_{\text{lowrank}}$ usually set to $(d_{\text{model}}/N)$.

During inference the controller uses a discrete argmax function, but during training the model uses a softmax to calculate and sample from a distribution. The model learns to select which row/column will be non-zero using the Gumbel-Softmax trick for discretization. To determine the active row/column in each block, we reparameterize sampling from a Bernoulli distribution by using the Gumbel-Softmax trick [25]. Instead of using the logits in each block to directly sample a binary value, we add independent noise from the Gumbel distribution to each of the logits, and then select the binary value with the highest logit (i.e., argmax) as the sample $z$. The argmax operation is not differentiable, but it can be approximated by a softmax with annealing temperature. Therefore, on the forward pass, we use the argmax to obtain a binary one-hot vector for each block, while on the backward pass, we approximate it with softmax. This approach is known as the Straight-Through Gumbel-Softmax estimator [14].

**Ablations.**    We investigate the impact of sparse FF on the model equivalent to T5-large with varying levels of sparsity, with $d_{\text{model}} = 1024$, $d_{\text{ff}} = 4096$, and 16 attention heads. When we set the sparsity level to $N$ (for e.g. $N = 64$) then every block of size $N$ has one non-zero value activated for inference. During training, the controller uses the bottleneck layer with $d_{\text{lowrank}} = 64$ and temperature of Gumbel softmax estimator set to 0.1. To improve training stability, the controller in the forward pass will use the output of argmax that is a binary one-hot vector for each block with a probability of 30% and otherwise it uses the output of softmax. Table 2 and Figure 3 show the perplexity and the decoding time of this model with varying levels of sparsity in feedforward layer. As the level of sparsity increases from 0 to 128, we observe a significant decrease in the decoding time, while the neg-log-perplexity of the model with $N = 64$ sparsity is comparable to the baseline.

| | Dec. time |
|---|---|
| baseline | 0.160s |
| Sparse FF 64 | 0.093s |
| Sparse FF 128 | 0.089s |

*Table 2: Decoding time of a singe to-ken decreases with increasing level of sparsity in the FF layer.*

*Figure 3: Log-perplexity of Scaling Transformers with Sparse Feedforward layer is very similar to dense base-line for sparsity level $N = 64$ but degrades slightly for N=128.*

We also checked the performance of the feedforward block with Mixture-of-Experts [33] style sparsity. As expected, this technique achieved decoding time comparable to sparse FF – 0.11s instead of 0.09s – but with its lack of granularity it achieved log-perplexity of 1.64, worse than both our method and the dense baseline.

## 3.2 Sparse QKV Layer

The decoding speed for a model with sparse feedforward blocks is dominated next by the query, key, value and output computation—the dense layers in attention, which we jointly call a QKV layer. Each of these dense layers has $d_{\text{model}}^2$ parameters and computation cost. Unfortunately, QKV layers don't have ReLUs, so the method used above to sparsify feedforward blocks is not viable here.

To make QKV layers sparse, we subdivide the dimensionality of the layer, $d_{\text{model}}$, into $S$ modules of size $M = d_{\text{model}}/S$, similar to splitting an activation vector into multiple heads. These modules can be processed with a convolutional layer with fewer weights and faster computation. However, with naïve design each module (and corresponding attention head) could access only a small part of a given token embedding. To alleviate that, we develop a multiplicative layer that can represent an arbitrary permutation and has fewer parameters and lower computation time than a dense layer. This multiplicative layer is inserted right before the convolutional layer, letting each head access any part of the embedding (see Figure 4(a)). This solution yields well-performing models that also decode fast.

**Multiplicative dense layer.** Our new multiplicative dense layer can represent an arbitrary permuta-tion and has $d_{\text{model}}^2/S + d_{\text{model}}S$ parameters, dependent on the sparsity hyperparameter $S$. It processes an input vector $\mathrm{x} \in \mathbb{R}^{d_{\text{model}}}$ by splitting it into S "modules" of size $M = d_{\text{model}}/S$. It produces output $\mathrm{y} \in \mathbb{R}^{S \times M}$ as follows

$$\mathrm{y}_{s,m} = \sum_i \mathrm{x}_i D_{i,s} E_{i,m}$$

where the two weight matrices are $D \in \mathbb{R}^{d_{\text{model}} \times S}$, and $E \in \mathbb{R}^{d_{\text{model}} \times M}$ (see Figure 4(b)). This layer executes significantly faster during inference because of the decreased number of parameters which need to be loaded from memory. Unless stated otherwise, we use $S = 16$.

The multiplicative layer is designed primarily to represent any permutation, so that each attention head can access information from any part of the embedding. We first verify that the multiplicative layer can indeed represent an arbitrary permutation (the proof is presented in the Appendix).

**Theorem 1.** *For any bijective function $f : \{1 \cdots d_{model}\} \Rightarrow \{1 \cdots S\} \times \{1 \cdots M\}$ there exists a pair of weights of multiplicative layer D, E such that $x_i = y_{s,m}$ for $\{s, m\} = f(i)$.*

**Convolutional layer.** The output of the multiplicative layer is a tensor of type/shape $\in \mathbb{R}^{\text{batch} \times \text{length} \times S \times M}$. We process this tensor with a two-dimensional convolutional layer, treating the length dimension and number of modules $S$ like height and width of an image. This layer uses $M$ filters and a kernel size of $F \times F$ so that each filter looks at $F$ modules ('S' axis) of the last $F$ tokens ('length' axis). Replacing the standard dense layer with such a convolution reduces the parameter

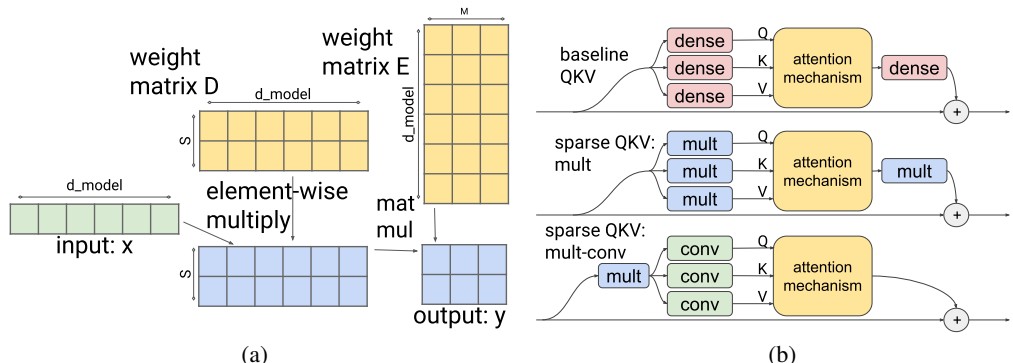

(a)                                                    (b)

*Figure 4: (a) Multiplicative layer can represent an arbitrary permutation, but has fewer parameters and reduced computation time compared to a dense layer. (b) Sparse QKV layer replaces Q, K, and V dense layers by composing multiplicative and convolutional layers and reducing the number of parameters and decoding time.*

count and computation time of the QKV layer. At the same time, by convolving over the 'length' axis, the model can incorporate more context into this computation [23].

The output of this layer has the same shape as the input. The optimal value of $S$ is less than $\sqrt{d_{\text{model}}}$. Empirically we set $F$ to $3$, $S$ equal to the number of heads in the attention mechanism and $M$ to be the dimensionality of a single attention head. In this case, we can feed the output of the convolution directly to the attention mechanism without reshaping the output. This convolutional layer has fewer parameters ($9M^2 + M = F^2(d_{\text{model}}/S)^2 + (d_{\text{model}}/S)$), and lower computational complexity ($O(d_{\text{model}}^2/S)$). Unless stated otherwise, we use $S = 16$ and $F = 3$.

**Combining multiplicative and convolutional layers.**    There are four dense layers to replace in the original attention mechanism: Q, K, V, and output. As shown in Figure 4(b), we replace Q, K, and V dense layers by composing multiplicative and convolutional layers, but with a multiplicative layer shared across all three: $Q = \text{conv}_Q(\text{mult}(x))$, $K = \text{conv}_K(\text{mult}(x))$, $V = \text{conv}_V(\text{mult}(x))$. We remove the output dense layer. Note that the combined multiplicative-convolutional variant has the output dense layer removed, while the other variants have it replaced with their respective sparse layers. Including this output layer negatively impacts decoding time. We can set the parameter $S$ to around $\sqrt{d_{model}}$, getting the number of layer parameters to scale proportionally to $d_{model}^{1.5}$ compared to $d_{model}^2$ of standard QKV layer.

**Interpretation of QKV layer.**    Note that when parameter $S$ in convolutional layer is equal to the number of heads in the attention mechanism, which is the case in our experiments, then each of the S modules corresponds to a single attention head. Therefore, the model uses the convolution to process each head using the same linear projection. Without the multiplicative layer this projection would operate on a predetermined part of the embedding layer for each head. However, by adding it the model can perform arbitrary permutation of dimensions, so each head can have access to arbitrary subset of embedding dimensions, not a predetermined subset of them. This fact helps with keeping the expressibility of resulting QKV layer despite the reduced number of parameters.

**Ablations.**    We investigate the impact of sparse QKV layers on the model equivalent to T5-large in Figure 5. We increase the value of $d_{\text{ff}}$ from 4096 to 6144 to preserve the number of parameters (see the next subsection for details). The decoding time with sparse QKV layer variants is similar to the baseline because it is dominated by the dense feedforward layer (details in appendix).

**Combined feedforward and QKV sparsity.**    Sparse QKV layers lower the total number of model parameters. To keep the model size matched to the baseline, we increase $d_{\text{ff}}$ to keep the number of parameters similar across all models we compare. For the T5-Large equivalent model, we increase $d_{\text{ff}}$ from 4096 to 6144. With increased $d_{\text{ff}}$, decoding time in the feedforward layer increases and thus, Sparse QKV layers alone do not speed up the model. However, when we combine Sparse QKV layers with sparse FF layers, we get a 3.05x speedup in decoding time of each decoding block with

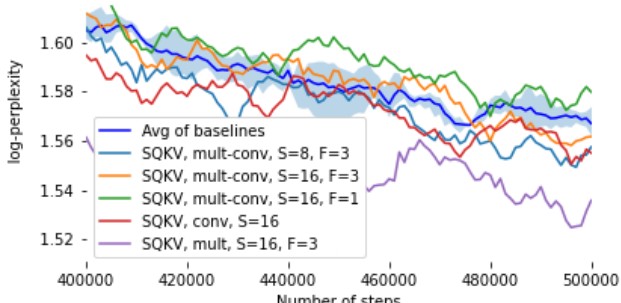

*Figure 5: Log-perplexity of Scaling  Transformers with Sparse QKV with different sparsity levels (S) and kernel sizes (F) is very similar to dense baseline within variance while multi-layer even improves perplexity.*

| | RTE | MRPC | SST-2 | QNLI | MNLI-m | QQP |
|---|---|---|---|---|---|---|
| Baseline Transformer (dense) | $70.1 \pm 1.1$ | $83.6 \pm 0.72$ | $92.6 \pm 0.85$ | $88.6 \pm 0.5$ | $78.5 \pm 0.41$ | $85.2 \pm 0.6$ |
| Scaling Transformer (Sparse FF+QKV) | 68.4 | 81.2 | 91.6 | 90.1 | 82.9 | 89.9 |
| Terraformer (Sparse FF+QKV) | 66.1 | 84.6 | 92.3 | 88.3 | 79.1 | 85.5 |

*Table 3: Accuracy of Scaling Transformer model and Terraformer model with sparse QKV+FF is comparable to the baseline Transformer within variance. The results are obtained by fine-tuning on selected downstream tasks from the GLUE dataset (validation split).*

comparable perplexity (see Table 1 and Figure 1). While the baseline these is a vanilla Transformer, the decoding speed is almost the same for a Reformer model as well.

Table 3 shows the accuracy of fine-tuning the model for downstream tasks from the GLUE dataset. Note that the model with sparseFF+QKV achieves accuracy similar to the baseline.

### 3.3   Sparse loss layer.

A final dense layer maps the model embedding into vocabulary size to compute the loss. We can sparsify this part of the model by replacing the dense layer with a multiplicative layer similar to previous sections; this speeds up decoding time but may degrade perplexity. The results are presented in appendix.

## 4   Sparsity for Long Sequences

The above gains from sparsifying the dense layers are encouraging, but we omitted one fundamental issue. When applied to longer sequences, the gains would effectively be lost, as the decoding time will be dominated by attention operations. Luckily, a number of methods have been proposed to solve this problem for Transformers, see [41] for a survey. We focus on the LSH (Locality-Sensitive Hashing) attention from Reformer [19] and show how to integrate this sparse attention mechanism, as well as recurrent blocks, into a Scaling  Transformer, yielding a *Terraformer*.

### 4.1   Architecture for Long Sequences

While integrating sparse attention layers into a Scaling Transformer, we notice that the architecture of the Transformer decoder block is suboptimal and can be redesigned to make a better use of these layers. In particular, separating decoder self-attention and encoder-decoder attention is not necessary any more from the perspective of efficiency. We therefore remove the encoder-decoder attention, but just concatenate the encoder representations before the decoder tokens. Doing this alone isn't enough though, since we took away one attention mechanism (encoder-decoder attention). We remedy this by having two attention mechanisms before the feedforward block. This simple architecture is as fast as the baseline Transformer while giving better results.

Putting this together, if $v_{enc}$ are the encoder activations and $v_{dec}$ are the decoder embeddings, the input to the decoder block $x$ is their concatenation on the length axis, LengthConcat($v_{enc}, v_{dec}$).

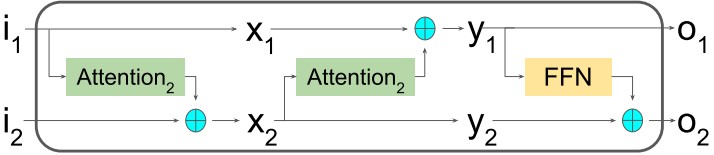

*Figure 6: Reversible decoder block in Terraformer.*

Each decoder block can be represented as:

$$y_1 = x + \text{Dropout}(\text{Attention}(\text{LayerNorm}(x)))$$
$$y_2 = y_1 + \text{Dropout}(\text{Attention}(\text{LayerNorm}(y_1)))$$
$$y = y_2 + \text{FFN}(y_2)$$

where $y$ becomes the input to the next decoder layer. See the appendix for a full diagram of the resulting architecture.

## 4.2 Reversibility for Memory Efficiency

To enable training Terraformer with large batches, and to fine-tune even large models on single machines, we apply ideas from the Reformer [19], in particular, reversible layers for the encoder and decoder blocks.

The original Reformer decoder block contained feedforward and attention layers in a 1-1 ratio. In the Terraformer architecture, as described above, there are two attention layers in the decoder block, so there are three swaps in the reversible layers in the decoder block (see Figure 6). In our experiments, this significantly improved performance.

Another issue with reversibility is that it is only formally correct for continuous functions. We find that this is not just a formal issue, but an important problem in practice. To make reversible layers train well with sparsity, we need to store the discrete decisions—i.e., the integers saying which rows to select—and use them for reversing. Recalculating these decisions on the backwards pass leads to worse results.

## 4.3 Recurrence for Generalization

In addition to incorporating sparse attention and reversibility, we also add recurrence to the feedforward block of Terraformer. Recurrent layers allow information to propagate in time, even in a single decoder block. It is challenging though to use them without decreasing model speed, esp. in training. For that reason, we use simple recurrent units [20] which parallelize well during training.

SRUs contain dense layers, so their use could negate the benefits of sparsity elsewhere. We tried a few methods to alleviate that, but it turns out that simply reducing the dimensionality of the SRUs works. So we first project from $d_{\text{model}}$ to a small dimension (32 in our experiments), then apply the SRU, and then project back to $d_{\text{model}}$ and add the result to the feedforward block. This low-rank recurrence is in our experiments sufficient to transfer enough information through time for the network to generalize.

Since the effects of SRUs on C4 are minimal (as the training and evaluation data are very similar), we use synthetic tasks to investigate out-of-distribution generalization. We train the models on long addition and on the task of copying a decimal digit. We train on inputs with at most 128 digits and evaluate on inputs lengths from 256 to 300, so over 2x longer. As can be seen in the table below, the baseline Transformer does not generalize well, while Terraformer manages to get a large portion correctly, even if it is not perfect like the Neural GPU [16].

## 4.4 Experiments

We designed Terraformer so that the benefits from sparsity would not be lost on long sequences, nor on downstream finetuning tasks. To test this, we chose the task of summarizing scientific papers

| Model | copy | copy (seq) | add | add (seq) |
|---|---|---|---|---|
| Transformer | 79.8% | 0% | 36.4% | 0% |
| Terraformer | 99.9% | 93.9% | 86.9% | 32.4% |

*Table 4: Comparison of out-of-distribution generalization for Terraformer and Transformer on two toy tasks, long addition and copying on decimal numbers. Under (seq) we report the number of fully correct sequences generated as answers.*

| Model | R-1 | R-2 | R-LSum | R-LSent |
|---|---|---|---|---|
| Terraformer | 45.40 | 17.86 | 41.21 | 26.33 |
| DANCER RUM | 42.70 | 16.54 | 38.44 | — |
| BIGBIRD-RoBERTa | 41.22 | 16.43 | 36.96 | — |
| Pegasus Large (C4) | 44.21 | 16.95 | 38.83 | 25.67 |
| DANCER PEGASUS | 45.01 | 17.6 | 40.56 | — |
| BIGBIRD-Pegasus | 46.63 | 19.02 | 41.77 | — |

*Table 5: Terraformer is competitive with strong baselines [46, 45, 10] on the ArXiv summarization task, without using the Pegasus loss and without beam search. On R-1, R-2 and R-LSum, Terraformer outperforms all previous models except for BigBird-Pegasus.*

using the dataset of scientific papers from arXiv[3][6]. In this task, the input is a whole paper—a long sequence—and the model is asked to output its abstract. Several recent papers studied this dataset and tasks and it has been shown [46, 45] that pretraining on C4 yields significant improvements on this task. We also pretrain Terraformer on C4 (like in all experiments in this paper) and fine-tuned it on the arXiv summarization task. We find that Terraformer is competitive with the above baselines, even though we mask single words (we do not use the Pegasus sentence loss) and decode the answers in a greedy way (no beam search). Note that ROUGE scores are computed using open-source scorer[4] with the metrics described in its documentation[5]. We also observe certain confusion between ROUGE-L metrics reported. As noted in the open-source scorer, there are two versions of ROUGEL-Sentence-Level (R-LSent) and ROUGEL-Summary-Level (R-LSum). For clarity, we report both of these metrics. Furthermore we only report the F1 measure of any ROUGE metric. We include a few examples of the generated abstracts in the appendix.

We pretrained Terraformer in the same way as all other baselines reported in this paper with the same number of parameters (800M), the same dimensions as mentioned before, and loss sparsity 4 to get the fastest model. Compared to the sparse Transformer model from the previous section that achieves a decoding speed of 0.061s, Terraformer achieves a decoding speed of 0.086s with a similar performance in terms of perplexity (see appendix for details). We also observe that the Terraformer model achieves accuracy similar to the Transformer model in Table 3 for selected downstream tasks on GLUE dataset.

Table 6 shows the speedup in decoding with sparse layers when we scale up Terraformer to 17B parameters. Note that sparsifying all the layers gives us 37x speedup in decoding.

## 5    Conclusion

When starting to investigate sparse variants of Transformers, we assumed that there would be a price to pay for sparsity—that a sparse model would always underperform a dense one with the same number of parameters. To our surprise, this is not the case: sparse is enough!

In our experiments with large models on the C4 dataset, the sparse models match the performance of their dense counterparts while being many times faster at inference. And, when scaling the models up, the benefits of sparsity become even larger. This promises to put Transformers back on a sustainable track and make large models more useful.

---

[3]We provide full details of our datasets, hyperparameters, and everything needed to reproduce the results in the appendix. The code is open-sourced as part of Trax 1.4.0 at `https://github.com/google/trax`.

[4]`https://pypi.org/project/rouge-score/`

[5]`https://github.com/google-research/google-research/tree/master/rouge`

| Terraformer | Dec. time | Speedup |
|---|---|---|
| dense | 3.651s | 1x |
| Sparse FF | 1.595s | 2.29x |
| SparseFF+QKV | 0.183s | 19.98x |
| SparseFF+QKV+loss | 0.097s | **37.64x** |

*Table 6: Decoding speed of a single token for Terraformer with 17B parameters is 37x faster than a dense baseline model, requiring less than 100ms/token for inference. Here attention-sparsity = 64, ff-sparsity = 256, and loss-sparsity = 4.*

The current results have a number of limitations. For one, the practical speedups we see are only for inference, not at training time. Moreover, we consider unbatched inference on CPUs, while often inference is ran in batched mode on GPUs. We believe with more work sparsity can bring improvements in these settings too, as our fundamental result shows that the sparse models reach the same perplexity as their dense counterparts with the same number of parameters.

So while we demonstrate that Scaling Transformers are possible, we consider this paper as a first step on the way to sustainable large models. There are numerous techniques for making models faster that could greatly benefit Terraformer and other Scaling Transformers. For example, we did not study quantization and we believe that it can make Scaling Transformers even faster. We also focused on inference speed and did not get improvements in training speed. The main reason is our use of Gumbel-Softmax when training the feedforward block (see Section 3.1). Fedus et al. [8] already provide a promising alternative, and we look forward to exploring it in future work.

Further, we hope that the community will take inspiration from Scaling Transformers and tune them for their needs. We ran experiments using layer sizes and hyperparameters borrowed from dense Transformers and they are most probably not optimal for Scaling Transformer. With proper tuning and further improvements we believe one could train a Scaling Transformer to match GPT-3 in accuracy but also run inference in reasonable time on a laptop. We put it as a fascinating challenge to the community, since such Scaling Transformers will not only be more sustainable but will also make large models accessible to everyone.

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
