## 6 Appendix

### 6.1 Sparse QKV

Sparse QKV uses a multiplicative layer to represent any permutation before composing this with a convolutional layer. We present the proof that this multiplicative layer can represent any permutation below.

**Theorem.** *With Multiplicative layer defined as*

$$y_{s,m} = \sum_i x_i D_{i,s} E_{i,m}$$

*For any bijective function $f : \{1 \cdots d_{model}\} \Rightarrow \{1 \cdots S\} \times \{1 \cdots M\}$ there exists a pair of weights of multiplicative layer D, E such that $x_i = y_{s,m}$ for $\{s, m\} = f(i)$.*

*Proof.* Let's take a function $f$, and define functions $s, m : s(i), m(i) = f(i)$. We construct weights $D_{i,s'} = (1 \text{ if } s' = s(i) \text{ otherwise } 0)$ and $E_{i,m'} = (1 \text{ if } m' = m(i) \text{ otherwise } 0)$. With those constraints we can derive, from the definition of multiplicative layer:

$$y_{s',m'} = \sum_i (x_i \text{ if } D_{i,s'} = 1 \wedge E_{i,m'} = 1 \text{ otherwise } 0)$$

$$y_{s',m'} = \sum_i (x_i \text{ if } s' = s(i) \wedge m' = m(i) \text{ otherwise } 0)$$

$$y_{s',m'} = \sum_i (x_i \text{ if } f(i) = s', m' \text{ otherwise } 0)$$

Because function $f$ is injective we can use its inversion.

$$y_{s',m'} = \sum_i (x_i \text{ if } i = f^{-1}(s', m') \text{ otherwise } 0)$$

$$y_{s',m'} = x_{f^{-1}(s',m')}$$

$$y_{f(i)} = x_i$$

$\square$

Figure 7: Log-perplexity of baselines and Scaling Transformers with just Sparse Loss, and varying number of modules.

### 6.2 Sparse Loss

To make the loss layer sparse, we investigate the impact of replacing the dense layer with the multiplicative layer designed for Sparse QKV layer. Table 7 and Figure 7 shows that increasing the sparsity of the loss layer degrades the perplexity slightly while speeding up the decoding time.

| Sparse loss | Dec. time |
|---|---|
| baseline | 0.160 s |
| S=2 | 0.158 s |
| S=4 | 0.149 s |
| S=8 | 0.148 s |

*Table 7: Decoding times by varying the number of modules $S$ in sparse loss layer.*

## 6.3 Sparsity Results Data

The results presented in Figure 1 are also accessible via a public Tensorboard link here `https://tensorboard.dev/experiment/on35sXCoTRSoI48ZomOnsw`

## 6.4 Architecture for Terraformer

Figure 8 shows the whole architecture of Terraformer model discussed in Section 4.1.

## 6.5 Pretrained Terraformer on C4 dataset

We pretrained Terraformer in the same way as all other baselines reported in this paper (see above), with one difference: we used 4x the batch size. (Thanks to reversibility, Terraformer can be trained with larger batches.) Table 8 shows the perplexity and decoding speed of the Terraformer model in comparison to the baseline Transformer model and the sparse Transformer model from the previous section. All models have the same number of parameters (800M) and the same dimensions as mentioned before. We used loss sparsity 4 for Terraformer to get the fastest model, so in Table 8 we compare it to a sparse Transformer with the same sparse loss.

|  | steps | batch size | Log perpl. | Dec. time |
|---|---|---|---|---|
| baseline Transf. | 500k | 4 | 1.57 | 0.160s |
| sparse Transf. | 500k | 4 | 1.61 | 0.061s |
| Terraf. | 125k | 16 | 1.66 | 0.086s |
| Terraf. | 150k | 16 | 1.63 | 0.086s |
| Terraf. | 175k | 16 | 1.59 | 0.086s |

*Table 8: Terraformer (800M) trained with 4x larger batch size achieves log-perplexity similar to baseline dense Transformer and Scaling Transformers with sparse FF+QKV and sparse loss. Terraformer trained with larger batch size does not match the perplexity of the baseline at $\frac{1}{4}$th number of steps, but catches up at around $\frac{1}{3}$rd—we believe this may be due to the fact that we used training hyperparameters optimized for the baselines. Decoding of a single token is 1.92x faster than baseline.*

## 6.6 Finetuning Terraformer on summarization task

We present a few examples of the abstracts generated by the Terraformer model for scientific papers in the ArXiv dataset [6]. Table 9 compares these abstracts to the corresponding examples from Tables I.25-27 in section I of [46].

The abstracts are decoded using greedy algorithm with temperature $T = 0.5$.

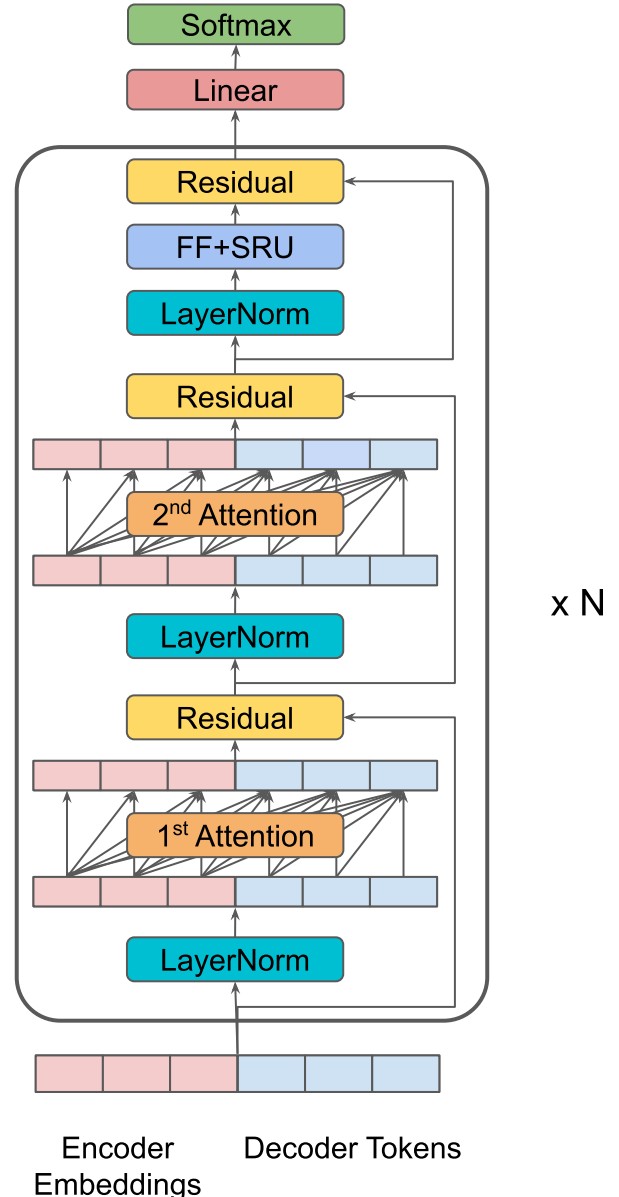

*Figure 8: Terraformer Architecture uses two attention mechanisms before the feedforward block in each decoder block.*

| | ArXiv |
|---|---|
| Document (ID #34) | consider a set of objects which should be ranked on the basis of information about their bilateral relationships . similar problems arise , among others , in social choice theory xcite , sports xcite , psychology xcite , internet search xcite , and bibliometrics xcite . we discuss a universal version of the problem involving arbitrary preference intensities as well as incomplete and multiple comparisons . the main contribution of this paper is the presentation of an impossibility theorem : consistency requiring that if an object is ranked at least as high as another in two independent problems , then it is ranked as high as the other in the unified problem , too and self - consistency a less known but intuitive property , introduced in xcite , which prohibits to assign a lower rank for an object with a better or equivalent performance than another can not be met simultaneously by any ranking method on the set of all problems . domain restrictions and weakening of the properties are also investigated in order to get some positive results . since self - consistency is an axiom difficult to debate , we can not demand consistency from a ranking method . at the first glance , it is a somewhat surprising and controversial result making the aggregation of ( incomplete ) preferences impossible . one may argue that the whole concept of paired comparisons - based ranking is flawed . however , loss of consistency may be regarded as a necessary sacrifice for dimension reduction since the outcome should be an inherently transitive ranking even from intransitive data . this interpretation is reinforced by the connection between consistency and independence of irrelevant matches ( which requires the relative ranking of two objects to be independent of the outcome of ... ... ... |
| Golden | assume a set of objects is given with information about their bilateral relationships , allowing for incomplete and multiple comparisons as well as different preference intensities . an axiomatic approach is applied for the problem of ranking the objects . consistency requires the preservation of relative ranking if two sets of such preferences are aggregated . self - consistency assigns the same rank for objects with the same performance , furthermore , an object should be ranked strictly higher if it shows an obviously better performance than another . it is revealed that these two properties can not be satisfied simultaneously . the impossibility holds under various restrictions on the domain . however , a positive result emerges if only the aggregation of ranking problems with the same comparison structure is allowed , for example , we have two round - robin tournaments . |
| Terraformer | we discuss a universal version of the problem involving arbitrary preference intensities as well as incomplete and multiple comparisons . the main contribution of this paper is the presentation of an impossibility theorem : consistency requiring that if an object is ranked at least as high as another in two independent problems , then it is ranked as high as the other in two independent problems , too and self - consistency a less known but intuitive property , introduced in @xcite . first , independence of irrelevant matches ( posing a kind of independence concerning the relative ranking of two objects ) is implied by three independent axioms , neutrality ( independence of relabelling the objects ) , symmetry ( implying a flat ranking if all comparisons are drawn ) , and consistency ( perhaps the most natural property of additivity ) . it turns out that consistency and self - consistency are still enough to get this result . the size of the problem does not solve the issue as it emerges with at least four objects . the size of the problem does not solve the issue as it emerges with at least four objects . the size of the problem is not a trivial extension of the aggregation of ( incomplete ) preferences . |
| ROUGE-1 | 41.87 |

| | |
|---|---|
| Document (ID #289) | machine learning methods are used widely within high energy physics ( hep ) . one promising approach , used extensively outside of hep for applications such as handwriting recognition , is that of support vector machines ( svms ) , a supervised learning model used with associated learning algorithms for multivariate analysis ( mva ) . developed originally in the 1960s , with the current standard version proposed in 1995 xcite , svms aim to classify data points using a maximal margin hyperplane mapped from a linear classification problem to a possibly infinite dimensional hyperspace . however this means svms , like other mva classifiers , have a number of free parameters which need to be tuned on a case by case basis . this motivates a number methods for ensuring the classifier is sufficiently generalised such that when used on an unseen dataset the performance can be accurately predicted . in this paper a brief overview of svms is given in section [ sec : svm ] , with an example using svms shown in section [ sec : checker ] . generalisation is discussed in section [ sec : gen ] with an illustrative example of how this can improve performance given in section [ sec : check2 ] . consider the problem of linear classification with the svm where the training set , xmath0 , is linearly separable . we define a separating hyperplane given by xmath1 , where xmath2 , the weight vector , is perpendicular to the hyperplane , and xmath3 , the bias , determines the distance of the hyperplane from the origin ( fig . ) . a decision function defined by xmath4 is used to positively and negatively classify xmath5 , the points in the training set . without further constraint the ... ... ... |
| Golden | we review the concept of support vector machines ( svms ) and discuss examples of their use . one of the benefits of svm algorithms , compared with neural networks and decision trees is that they can be less susceptible to over fitting than those other algorithms are to over training . this issue is related to the generalisation of a multivariate algorithm ( mva ) ; a problem that has often been overlooked in particle physics . we discuss cross validation and how this can be used to improve the generalisation of a mva in the context of high energy physics analyses . the examples presented use the toolkit for multivariate analysis ( tmva ) based on root and describe our improvements to the svm functionality and new tools introduced for cross validation within this framework . |
| Terraformer | support vector machines ( svms ) are used widely within high energy physics ( hep ) for applications such as handwriting recognition . the svm algorithm is a supervised learning model used with associated learning algorithms for multivariate analysis ( mva ) . the hard margin svm is an important svm algorithm , and the hard margin svm is an important svm algorithm . in this paper a brief overview of svms is presented , with an example showing similar performance to that of a bdt . however it is not clear without further checks as to whether the mvas are sufficiently generalised . hence a multistage cross - validation procedure has been outlined , which for the same example shows better performance as well as better agreement between the training and testing samples in the output distributions . |
| ROUGE-1 | 43.54 |

| Document (ID #294) | the pure quantum states are traditionally associated with the wave function xcite or a vector in the hilbert space xcite . the mixed quantum states are described by the density matrix xcite or the density operator xcite . there exist several representations of quantum states in terms of the quasidistribution functions like the wigner function xcite and the husimi kano function xcite . the diagonal representation of quantum states was suggested in xcite ( see also xcite ) . it was studied and applied in xcite . in this representation , a quantum state is represented in terms of weighted sum of coherent - state $xmath0$ projectors . the properties of all the quantum - state representations considered are associated with the properties of the density operator which is hermitian , trace - class nonnegative operator . this means , in particular , that all the eigenvalues of the density operators must be nonnegative . in the quantum domain , the multipartite systems have a specific property connected with strong correlations of the quantum subsystems . this property provides the entanglement phenomenon xcite . in the diagonal representation of the density states , the weight function $xmath1$ is an analog of the probability - distribution function in the phase space . for some class of states , this function is identical to the probability - distribution function like in classical statistical mechanics . in xcite , the tomographic - probability representation of quantum states , where the quantum state is associated with the so - called symplectic tomogram , was introduced . the tomogram is a fair probability distribution containing the same information on quantum state that the density operator does ( or such its characteristics as the wigner or husimi kano functions ) . the aim of this work ... ... ... |
|---|---|
| Golden | the quasidistributions corresponding to the diagonal representation of quantum states are discussed within the framework of operator - symbol construction . the tomographic - probability distribution describing the quantum state in the probability representation of quantum mechanics is reviewed . the connection of the diagonal and probability representations is discussed . the superposition rule is considered in terms of the density - operator symbols . the separability and entanglement properties of multipartite quantum systems are formulated as the properties of the density - operator symbols of the system states . |
| Terraformer | the diagonal and probability representations of quantum states are reviewed . the superposition rule is formulated in both the diagonal and probability representations . the problem of separability and entanglement is formulated in both the diagonal and probability representations . the problem of separability and entanglement is formulated in both the diagonal and probability representations . the problem of separability and entanglement is formulated in both the diagonal and probability representations . the separability and entanglement properties of composed system in the tomographic and diagonal representations are analyzed . |
| ROUGE-2 | 27.16 |

| | |
|---|---|
| Document (ID #32) | deep neural networks ( dnns ) have been receiving ubiquitous success in wide applications , ranging from computer vision xcite , to speech recognition xcite , natural language processing xcite , and domain adaptation xcite . as the sizes of data mount up , people usually have to increase the number of parameters in dnns so as to absorb the vast volume of supervision . high performance computing techniques are investigated to speed up dnn training , concerning optimization algorithms , parallel synchronisations on clusters w / o gpus , and stochastic binarization / ternarization , etc xcite . on the other hand the memory and energy consumption is usually , if not always , constrained in industrial applications xcite . for instance , for commercial search engines ( e.g. , google and baidu ) and recommendation systems ( e.g. , netflix and youtube ) , the ratio between the increased model size and the improved performance should be considered given limited online resources . compressing the model size becomes more important for applications on mobile and embedded devices xcite . having dnns running on mobile apps owns many great features such as better privacy , less network bandwidth and real time processing . however , the energy consumption of battery - constrained mobile devices is usually dominated by memory access , which would be greatly saved if a dnn model can fit in on - chip storage rather than dram storage ( c.f . xcite for details ) . a recent trend of studies are thus motivated to focus on compressing the size of dnns while mostly keeping their predictive performance xcite . with different intuitions , there are mainly two types of dnn compression methods , which could be used in conjunction for better parameter savings . the ... ... ... |
| Golden | as the complexity of deep neural networks ( dnns ) trend to grow to absorb the increasing sizes of data , memory and energy consumption has been receiving more and more attentions for industrial applications , especially on mobile devices . this paper presents a novel structure based on functional hashing to compress dnns , namely funhashnn . for each entry in a deep net , funhashnn uses multiple low - cost hash functions to fetch values in the compression space , and then employs a small reconstruction network to recover that entry . the reconstruction network is plugged into the whole network and trained jointly . funhashnn includes the recently proposed hashednets xcite as a degenerated case , and benefits from larger value capacity and less reconstruction loss . we further discuss extensions with dual space hashing and multi - hops . on several benchmark datasets , funhashnn demonstrates high compression ratios with little loss on prediction accuracy . |
| Terraformer | this paper presents a novel approach to relieve the hash collision effect through a sparse feature space . specifically , we use multiple hash functions to map per virtual entry into multiple values in compression space . then an additional network plays in a _ mapping function _ role from these hashednets to the virtual entry before hashing , which can be also regarded as ž047 reconstructing " the virtual entry from its multiple hashed values . plugged into and jointly trained within the original network , the reconstruction network is of a comparably ignorable size , i.e. , at low memory cost . this functional hashing structure includes hashednets as a degenerated special case , and facilitates less value collisions and better value reconstruction . experiments on several datasets demonstrate promisingly larger reduction of model sizes and/or less loss on prediction accuracy , compared with hashednets . |
| ROUGE-2 | 16.11 |

| | |
|---|---|
| Document (ID #248) | stripped supernovae ( sne ) and long - duration gamma - ray bursts ( long grbs ) are nature s most powerful explosions from massive stars . they energize and enrich the interstellar medium , and , like beacons , they are visible over large cosmological distances . however , the mass and metallicity range of their progenitors is not known , nor the detailed physics of the explosion ( see reviews by xcite and xcite ) . stripped - envelope sne ( i.e , sne of types iib , ib , and ic , e.g. , xcite ) are core - collapse events whose massive progenitors have been stripped of progressively larger amounts of their outermost h and he envelopes ( fig . [ fig1 ] ) . in particular , broad - lined sne ic ( sne ic - bl ) are sne ic whose line widths approach 20,000xmath030,000 xmath1 around maximum light ( see below ) and whose optical spectra show no trace of h and he . for the last 15 years , the exciting connection between long grbs and sne ic - bl , the only type of sne observed accompanying long grbs ( for reviews , see xcite ) , and the existence of many more sne ic - bl without grbs raises the question of what distinguishes sn - grb progenitors from those of ordinary sne ic - bl without grbs . viewing angle effects are probably not the reason why those sne ic - bl did not show an accompanied grbs xcite and based the same radio upper - limits , only xmath2 1% of sne ib / c appear to be accompanied by grbs xcite . one promising line of attack is to investigate what sets apart sne ib ... ... ... |
| Golden | while the connection between long gamma - ray bursts ( grbs ) and type ib / c supernovae ( sne ib / c ) from stripped stars has been well - established , one key outstanding question is what conditions and factors lead to each kind of explosion in massive stripped stars . one promising line of attack is to investigate what sets apart sne ib / c * with * grbs from those * without * grbs . here , i briefly present two observational studies that probe the sn properties and the environmental metallicities of sne ib / c ( specifically broad - lined sne ic ) with and without grbs . i present an analysis of expansion velocities based on published spectra and on the homogeneous spectroscopic cfa data set of over 70 sne of types iib , ib , ic and ic - bl , which triples the world supply of well - observed stripped sne . moreover , i demonstrate that a meta - analysis of the three published sn ib / c metallicity data sets , when including only values at the sn positions to probe natal oxygen abundances , indicates at very high significance that indeed sne ic erupt |
| Terraformer | the discovery of stripped supernovae ( sne ) and long - duration gamma - ray bursts ( grbs ) , and the discovery of long - duration gamma - ray bursts ( grbs ) , has raised the question of what distinguishes sne ic - bl from those of ordinary sne ic - bl without grbs . here i present a meta - analysis of published metallicities at the explosion site of stripped supernovae with and without grbs , and a meta - analysis of published metallicities at the explosion site of sne ib / c with and without grbs . the results of our analysis are compared with those of sne ic - bl without grbs , and with those of sne ic - bl without grbs . the results of our study are compared with those of sne ic - bl without grbs , and with sne ic - bl with grbs . |
| ROUGE-LSent | 28.83 |

| | |
|---|---|
| Document (ID #270) | the alice experiment is dedicated to the study of the properties of qcd matter created in nucleus - nucleus collisions at lhc energies xcite . the inner tracking system in the alice apparatus is made of position sensitive detectors which have to handle several thousands tracks per unit of rapidity . the two innermost layers at 3.9 xmath10 and 7.6 xmath10 radii , respectively , constitute the silicon pixel detector ( spd ) . the spatial precision and hit efficiency of the spd are key parameters since they determine the alice capability of detecting particles with open heavy - flavour xcite . + the basic detector unit of the alice spd is the ladder , a two - dimensional silicon matrix of pxmath11n reverse biased diodes of dimensions 50 x 425 xmath12 , flip - chip bonded to five read - out chips . each diode is connected to a cell of the front - end read - out asic via a pb - sn solder bump of 25 xmath13 diameter . the detector contains nearly 10xmath14 active cells in total . the read - out is binary . to reduce the material budget , the sensor thickness is limited to 200 xmath13 and the read - out chip wafers are thinned down to 150 xmath13 . further details can be found in xcite . + early prototypes of the alice spd elements , in the form of single - chip assemblies , were tested in high energy proton / pion beams at the cern sps in 2002 and 2003 . these assemblies were made with sensors of 200 xmath13 and 300 xmath13 thicknesses , while the read - out chips ( unthinned ) were 725 xmath13 thick . those beam tests were primarily aimed at evaluating the performance of ... ... ... |
| Golden | the two innermost layers of the alice inner tracking system are instrumented with silicon pixel detectors . single chip assembly prototypes of the alice pixels have been tested in high energy particle beams at the cern sps . detection efficiency and spatial precision have been studied as a function of the threshold and the track incidence angle . the experimental method , data analysis and main results are presented . d. elia@xmath0 , g. anelli@xmath1 , f. antinori@xmath2 , a. badal@xmath3 , g.e . bruno@xmath4 , m. burns@xmath1 , i.a . cali@xmath5 , m. campbell@xmath1 , m. caselle@xmath4 , s. ceresa@xmath1 , p. chochula@xmath1 , m. cinausero@xmath6 , j. conrad@xmath1 , r. dima@xmath2 , d. fabris@xmath2 , r.a . fini@xmath4 , e. fioretto@xmath6 , s. kapusta@xmath1 , a. kluge@xmath1 , m. krivda@xmath7 , v. lenti@xmath4 , f. librizzi@xmath3 , m. lunardon@xmath2 , v. manzari@xmath4 , m. morel@xmath1 , s. moretto@xmath2 , p. nilsson@xmath1 , f. osmic@xmath1 , g.s . pappalardo@xmath3 , v. paticchio@xmath4 , a. pepato@xmath2 , g. prete@xmath6 , a. pulvirenti@xmath3 , p. riedler@xmath1 , f. riggi@xmath3 , l. sndor@xmath7 , r. santoro@xmath4 , f. scarlassara@xmath2 , g. segato@xmath2 , f. soramel@xmath8 , g. stefanini@xmath1 , c. torcato de matos@xmath1 , r. turrisi@xmath2 , l. vannucci@xmath6 , g. viesti@xmath2 , t. virgili@xmath9 spatial precision , efficieny , silicon pixel detector , alice , lhc 29.40.gx , 29.40.wk |
| Terraformer | the performance of prototype assemblies for the alice silicon pixel detector have been extensively studied by using beam test data collected in the past years at the cern sps . the cluster pattern distribution , the intrinsic spatial precision and the detection efficiency have been investigated as a function of both the detector threshold and the incident angle of the tracks . the results show a very high detection efficiency ( above 99% ) in a wide threshold range and a spatial precision of about 10 @xmath0 in the short pixel side coordinate for normal track incidence and 210 dac threshold . the detector performance with angled tracks has also been investigated . |
| ROUGE-LSent | 23.10 |

*Table 9: Abstracts generated by the Terraformer model on a set of examples of scientific papers from ArXiv. Note that the corresponding abstracts and scores for the Pegasus model are in Tables I.25-27 in section I of [46].*