# OpenReview forum: "Sparse is Enough in Scaling Transformers"
_NeurIPS.cc/2021/Conference — NeurIPS 2021 Poster_

### Official Review · Reviewer_ju7M · 2021-07-16

**Rating:** 7
**Confidence:** 3

**Summary:**

This work focuses on improving Transformer inference efficiency by redesigning the aforementioned architecture such that the matrix multiplications can be sparse at inference time. This sparsity is introduced in the dense layers that compute the query, key, and value, and in the feed-forward layer as well. This results in a much faster decoding speed when compared to dense transformers.

**Limitations And Societal Impact:**

The potential negative societal impact is adequately addressed.

**Main Review:**

_Originality_: The methods proposed are novel, and are clearly distinguished from related work.

_Quality_: The changes to the Transformer architecture that are proposed in this paper are interesting and very useful. They allow for much faster decoding. However, it's not clear if there are differences in training time due to additional blocks, like the "Controller". Can the authors clarify this? Furthermore, I would like to see the proposed models compared to more baselines, for example, the Reformer architecture (in performance and decoding time).

_Clarity_: The paper is mostly well-written, although I found it a bit confusing at times, particularly in §3. For example, are C_1 and C_2 additional parameter matrices? The method described in §3.2 also deserves a more thorough explanation.

_Significance_: As mentioned, this work proposes a Transformer variant in which the decoding speed is greatly increased, at little cost in performance. This result is very useful for practioners and researchers with varying ammount of computing resources, making it a valuable contribution to the community. The authors mention in the conclusion that the proposed technique can help to fine-tune a large Transformer "a single workstation". Can the authors clarify? Isn't your method only faster at inference time?

**Time Spent Reviewing:**

6h

---

> ### Author Response · Authors · 2021-08-10
> **Review Response**
>
> * Thank you for requesting clarification on training time speedup. The proposed sparse FF and QKV layers achieve 20x speed up in decoding time of unbatched sequences for 17B param model. We do not claim a training speedup with the proposed techniques as shown in the training curves of all techniques in the [Tensorboard link](https://tensorboard.dev/experiment/on35sXCoTRSoI48ZomOnsw)
>  in the appendix and we will emphasize this in the final version.
> * Thank you for the request for additional baselines concerning the Reformer architecture. In experiments that we ran, the decoding time of the Reformer is very similar (slightly higher) to that of the Transformer, while the proposed techniques are much faster. We will include these results in the final version of the paper.
> * We apologize for the confusion regarding the proposed techniques helping to finetune a large Transformer model using "a single workstation." The methods in the paper only speed up the decoding time, and the "finetuning on a single workstation" is not a contribution of the paper - just a feature of the code that we will open-source with the final version of the paper.

---

> > ### Comment · Reviewer_ju7M · 2021-08-13
> > **Response to rebuttal**
> >
> > Thank you for your clarifications. One of my questions still remains, though. Does the addition of the Controller module increase training time? Or is it negligible?

---

> > > ### Author Response · Authors · 2021-08-13
> > > **Controller training time clarification**
> > >
> > > Thank you for your question! The additional training time added by the controller is small because the controller is low-rank. We will measure this precisely and add a table with training time influence per component in the final version of the paper.

---

> > > > ### Comment · Reviewer_ju7M · 2021-08-14
> > > > **Response to rebuttal**
> > > >
> > > > Thank you. My concerns were addressed. I updated my score.

---

### Official Review · Reviewer_vUEu · 2021-07-18

**Rating:** 5
**Confidence:** 4

**Summary:**

The paper proposes a variant of the Transformer layer that includes sparsified dense and attention layers. For FFN layers, the author introduces a controller layer that selects only a subset of hidden dimensions to be used in the computation. For the attention layer, The author proposes a multiplicative transformation to simulate the permutation of the input hidden states and applies a convolution layer on top of it. The author shows no accuracy drop with the proposed method and speedup in the decoding phase.



**Limitations And Societal Impact:**

Please refer to the weaknesses in main review section.

**Main Review:**

Overall, I believe the proposed method in the paper is practically useful and can speed up the Transformers. However, I still have some doubts about the motivation and rationale of the proposed methods. Detailed comments:

Strengths
- The paper is clear and easy to follow. I can understand the proposed method very easily.
- The author did thorough experiments to verify the effectiveness of the method, including end-to-end experiments on multiple datasets and ablation studies of each method.

Weaknesses
- The method used for the FFN layer is similar to the previous mixture-of-expert method. The argmax selection (i.e. controller) part in the proposed method can be analogized to the expert selection part in mixture-of-expert. The difference that the authors mentioned in the related work doesn’t convince me: In mixture-of-expert, all of the experts are also being trained and only a subset of experts will be activated during the decoding for each token.
- The convolution layer in the proposed attention layer doesn’t have any semantic meaning. The convolution over the sequence length dimension can be understood because neighboring words in a sentence often have a more semantic correlation. But the convolution over the S dimension doesn’t have this kind of semantic meaning. Please elaborate on the rationale for the convolution design choice here. In addition, the sequence length and S dimension are very different, why is the convolution window size for both dimensions set to be F?
I would like to see more results on the speedup. What’s the speedup on the training speed? What’s the speedup with inference on a sequence or a batch of sequences?

Other questions:
- I haven’t seen people use negative log perplexity before. Why don't use log perplexity or just perplexity?



**Time Spent Reviewing:**

3

---

> ### Author Response · Authors · 2021-08-10
> **Review Response**
>
> * Regarding the sparse FF layer being similar to the previous mixture-of-expert method, note that the mixture-of-expert method divides the activation vector into N blocks and chooses one of those blocks to be non-zero, while the proposed sparse FF layer divides the activation vector into blocks of N vectors and chooses a single vector out of every block. Empirically, we found our technique performs better than Mixture of Experts in terms of perplexity (4.9 vs 5.2) and slightly better in decoding time (0.09s vs 0.11s). We will add these results to the final version of the paper.
> * Regarding the semantic meaning of the convolution layer in the sparse QKV, note that each of the S modules corresponds to a single attention head if S is set equal to the number of heads, which is the case in our experiments. Thus the model uses the convolution to process each head using the same linear projection, just with different data coming from different parts of the embedding or shuffled by the multiplicative layer. The exact window sizes for the convolutions do not influence the final results very much, and we will add ablations over these sizes and a better explanation to the final version of the paper.
> * The proposed techniques are targeting speedup in decoding time in the unbatched setting. The appendix contains a [link](https://tensorboard.dev/experiment/on35sXCoTRSoI48ZomOnsw) showing training curves of all the techniques. We do not claim a training speedup with the proposed methods and will emphasize this in the final version.
> * We apologize for the confusion caused by negative log-perplexity. We will change the y-axis on plots to show log-perplexity in the final version of the paper.

---

> > ### Comment · Reviewer_vUEu · 2021-08-24
> > **Response to rebuttal**
> >
> > Thanks for the response! The rebuttal partially addressed my concerns, but some of my concerns still remain so I will keep my original score:
> > - The authors claim that *”the proposed sparse FF layer divides the activation vector into blocks of N vectors and chooses a single vector out of every block”*. However, the selection of each block is highly correlated since the selection is decided by a low-rank compressed version of the activation. In other words, if the activation choose a specific vector for one block, one can predict with high confidence about what the vector selection be for another block. Therefore, I feel the expressiveness of the model doesn’t change much compared to the mixture-of-experts. The empirical results are helpful, but I’m not sure about whether a 0.3 points improvement on perplexity is significant.
> > - I understand the meaning of performing convolution on the S dimension now. However, I still feel like setting a convolution filter size on the S dimension is meaning less and the FxF 2D convolution filter seems arbitrary.

---

> > > ### Author Response · Authors · 2021-08-25
> > > **Clarifying the motivation for low-rank controller**
> > >
> > > We want to thank the reviewer for the insightful comments and a lot of thought put into our paper. In particular, we believe this is the key doubt, as stated by the reviewer: *"However, the selection of each block is highly correlated since the selection is decided by a low-rank compressed version of the activation."* We believe that the selections of blocks are not necessarily correlated as long as $d_\text{lowrank}$ is on the order of $\sqrt{d_\text{ff}}$ - as we propose in the paper.
> > >
> > > Let us analyze the amount of information both in the low-rank vector and in the choices that depend on it. The vector consists of $d_\text{lowrank}$ (in our experiments $d_\text{lowrank}$ is 32 or 64) activations, each a 32-bit number but effectively carrying less information. For an easy approximation let's assume it has 10 bits of information, for a total of 320 or 640 bits in the low-rank vector (for $d_\text{lowrank}$=32 or 64 respectively). On the other hand, with $d_\text{ff}$ = 4096 divided into 64 blocks, the controller needs to pick 1 in 64 vectors from each block. The information content here is 64 * $\log_2$(64), so 64*6 = 384 bits - very similar. So, there is enough capacity in the low-rank vector that there does not need to be any correlation between the block choices as long as $d_\text{lowrank}$ is on the order of $\sqrt{d_\text{ff}}$. We want to thank the reviewer for raising this issue and we will make sure to highlight and clarify this point in the final version of the paper.

---

### Official Review · Reviewer_BocG · 2021-07-19

**Rating:** 8
**Confidence:** 4

**Summary:**

Update: I've moved my score from 7 to 8.

The paper first introduces a sparse feedforward layer, which uses the input at every timestep to decide how to zero out the hidden vector (the one right in the middle) of the MLP feedforward layer. It uses the gumbel-softmax trick and zeros out (N-1) out of every N activations. This zeroing out of each hidden vector allows us to not use some of the rows & column of the W_1 and W_2 matrices of the feedforward layer.

The next method is the sparse QKV layer, where each input is compressed into a few smaller vectors that each do attention independently. The authors show that a convolutional layer can make this faster, without loss of accuracy, and also toss away the final output projection from their self-attention module.

The authors then apply a few off-the-shelf tricks to their transformer, such as reversible layers and replacing the enc/dec attention by attending to the concatenation of the enc/dec words.

The empirical results are very strong and show good performance on challenging tasks such as language modeling, summarization, and GLUE, while massively improving inference speed (training speed is not improved).



**Ethical Concerns:**

None.

**Limitations And Societal Impact:**

Yes.

**Main Review:**

Strengths:
The paper is well written, and the ideas presented are relatively simple to implement.
The paper presents strong empirical results on hard tasks, with big inference speedups.

Weaknesses:
I might be wrong about this, and I would be grateful if the authors corrected me in the author response period, but I believe that the methods shown in this paper mostly improve the speed of *unbatched* inference. If this is true than the authors should point it out in the paper, and if not then they should also show speed number of batched inference.
If my assumption is correct it might slightly weaken the results of the paper, since for most models I believe batched inference is used in production. It's still a strong paper that I believe should be accepted, but this limitation should be pointed out, and maybe future papers will be able to address it.


Missing Citations:
Line 222: "We therefore remove the encoder-decoder attention, but just concatenate the encoder representations before the decoder tokens". Authors must cite "Layer-Wise Coordination between Encoder and Decoder for Neural Machine Translation", He et al, which did exactly that.

Nitpicks:
Line 27/100/and throughout the paper: "output layers"- consider referring to these as the 'output projection'. I feel like calling them "output layers" might confuse some readers.
Line 103: " decoding speed is dominated by the execution cost of the feedforward block". Clarify that this is true only for shorter input sequences (the authors do mention this later in the paper but I feel it's important to also mention it in line 103).
Table 2: Consider adding the final perplexity of each model to the table (I know you also have it in the figure but it would be nice to have it in the table).

**Time Spent Reviewing:**

2

---

> ### Author Response · Authors · 2021-08-10
> **Review Response**
>
> * Thank you for requesting the clarification on batched vs unbatched inference. The proposed methods shown in this paper indeed speed up unbatched inference and the gains for batched inference will be smaller. We would like to emphasize that the proposed approach yields 20x improvement for a model with 17B parameters at batch size 1. Thus, we would still expect a few times improvement at higher batch sizes.
> * We thank the reviewer for the additional reference: "Layer-Wise Coordination between Encoder and Decoder for Neural Machine Translation" which we will include in the final version.
> * We also thank the reviewer for the detailed comments to improve the presentation of the paper and will address them in the final version of the paper.

---

> > ### Comment · Reviewer_BocG · 2021-08-21
> > **Thank you for addressing my concerns.**
> >
> > I do believe your improvements for unbatched inference are worthy of acceptance, I just hope that this is made clear in the final paper.

---

### Official Review · Reviewer_zmH4 · 2021-07-22

**Rating:** 6
**Confidence:** 4

**Summary:**

The paper presents an approach to training large-scale sparse transformers by sparsifying the activations of linear layers in the network (feedforward and Q, K, V projections). The approach is particularly appealing because it doesn't appear to require any hardware-specific or low-level optimizations to be efficient since it sparsifies activations. A controller network produces a block-wise one-hot mask on the activations and weights for the subsequent layer are selected on the fly based on this mask. Gradients are propagated through this hard-selection operation via the straight-through Gumbel softmax estimator.

The authors are able to achieve fairly substantial inference time speedups via sparsification. They also apply the same approaches to sparsifying models that are tailored towards handling long sequences such as the Reformer with LSH attention.

**Limitations And Societal Impact:**

The authors do not discuss the limitations of their work and its potential negative societal impact. There is plenty of prior literature to engage with here since there has been plenty of discussions recently on the societal impact of large language models.

**Main Review:**

Strengths:

1. The sparsification technique is simple and easy to implement and seems to work at scale.
2. The approach shows empirical results on multiple tasks and large-scale benchmarks.
3. Significant improvements in inference times.
4. The paper is well written and easy to understand.

Weaknesses:

1. The paper seems to lack comparisons to more traditional weight pruning-like strategies which also promise minimal loss in task performance while being faster at inference.
2. It would be interesting to see ablations that push for slightly stronger sparsity constraints and how it affects task performance.
3. While the proposed method doesn't require hardware-specific or low-level library optimizations, it appears that block sparsity or other kinds of structured sparsity may be optimized on specialized hardware in the near future such as in https://developer.nvidia.com/blog/exploiting-ampere-structured-sparsity-with-cusparselt/. A discussion about the compatibility of the kinds of block-wise activation sparsity used in this work and structured weight sparsity would be useful.

Overall comments:

- I'd encourage the authors to rethink the title of the paper since it isn't clear what "Enough" is in the context of model scaling or even along what axis you are scaling models. The title is potentially misleading into having readers think that it enables efficient training, while this is only helps at inference.


**Time Spent Reviewing:**

5

---

> ### Author Response · Authors · 2021-08-10
> **Review Response**
>
> We are very grateful for all the comments and the work the reviewer put into them, as it allows us to improve our paper significantly. Below, we clarify a few issues that were raised and which will be improved in the final version of the article.
>
> * Regarding comparisons to weight pruning, we would like to emphasize that it is complementary to the improvements presented in this paper. We believe the techniques presented in the paper can be combined with weight pruning. Moreover, our approach to sparsity yields over 2.6x speedup in wall-clock decoding time for a model with 800M parameters and 20x improvement for a model with 17B parameters, which is more than the usual weight pruning strategies.
> * Regarding ablations with stronger sparsity constraints, we ran additional experiments where the sparse feedforward layer is forced to have a mixture of experts style block sparsity constraint resulting in higher perplexity (5.2 vs 4.9) compared to the proposed approach. We will include these results in the final version.
> * Regarding compatibility with hardware-specific optimizations, we will add a discussion about the compatibility of the proposed sparse feedforward layer with hardware that enables structured weight sparsity and potential gains there. We expect that our proposed techniques will give speedup on any hardware that can speed up the loading of block sparse matrices.
> * Thank you for the feedback regarding the title of the paper. We will modify the title to clarify that the proposed sparsity addresses decoding time speedups.

---

> > ### Comment · Reviewer_BocG · 2021-08-21
> > **I agree with the authors**
> >
> > I agree with the author's assertion that weight pruning is orthogonal to their proposal and that the two could be combined. I do not believe that not comparing to weight pruning strategies should be seen as a weakness of the work.

---

> > ### Comment · Reviewer_zmH4 · 2021-08-24
> > **Thanks for the clarifications**
> >
> > Thank you for clarifying some of my comments on ablations for stronger sparsity constraints and weight pruning. I agree that weight pruning can be combined with your approach and but still think there is merit to having it to compare inference-only weight vs block-activation sparsity.

---

### Decision · Program_Chairs · 2021-09-27

**Decision:**

Accept (Poster)

**Comment:**

The paper introduces a technique for improving the inference speed of transformers by sparsifying all the linear layers. During training, a controller module predicts a sparse mask for the activations (trained with a straight-through Gumbel softmax), and during decoding this mask is used to prune the weights for the linear layer. Reviewers appreciated that the method is simple, and dramatically improves the speed of decoding. Reviewer vUEu raises reasonable concerns about the lack of comparison with mixture-of-experts approaches (which improve both training and inference speed), however the authors argue that their method is more expressive, and include an empirical comparison in their response. I think the title and abstract should be clearer about that the paper is addressing purely unbatched decoding speed. Overall, I recommend acceptance.